# Site-Specific Metastasis and Survival in Papillary Thyroid Cancer: The Importance of Brain and Multi-Organ Disease

**DOI:** 10.3390/cancers13071625

**Published:** 2021-04-01

**Authors:** Eman A. Toraih, Mohammad H. Hussein, Mourad Zerfaoui, Abdallah S. Attia, Assem Marzouk Ellythy, Arwa Mostafa, Emmanuelle M. L. Ruiz, Mohamed Ahmed Shama, Jonathon O. Russell, Gregory W. Randolph, Emad Kandil

**Affiliations:** 1Department of Surgery, Tulane University School of Medicine, New Orleans, LA 70112, USA; mhussein1@tulane.edu (M.H.H.); mzerfaoui@tulane.edu (M.Z.); aattia@tulane.edu (A.S.A.); remmanuelle@tulane.edu (E.M.L.R.); mshama@tulane.edu (M.A.S.); 2Genetics Unit, Department of Histology and Cell Biology, Suez Canal University, Ismailia 41522, Egypt; 3Tulane University School of Medicine, New Orleans, LA 70112, USA; aellythy@tulane.edu (A.M.E.); arwam9@gmail.com (A.M.); 4Division of Head and Neck Endocrine Surgery, Department of Otolaryngology-Head and Neck Surgery, Johns Hopkins, Baltimore, MD 21287, USA; jrusse41@jhmi.edu; 5Division of Thyroid and Parathyroid Endocrine Surgery, Department of Otolaryngology, Massachusetts Eye and Ear Infirmary and Harvard Medical School, Boston, MA 02115, USA; gregory_randolph@meei.harvard.edu

**Keywords:** metastasis, survival, thyroid cancer, multiple organ metastasis

## Abstract

**Simple Summary:**

Papillary thyroid cancer (PTC) is the most common subtypes of thyroid malignancy, and its distant metastasis (DM) is linked with higher mortality. We sought to study the consequences of different distant metastasis sites on the survival of PTC patients to better understand their association with survival outcomes, which will help clinicians to develop tailored treatment plans. Our results showed that metastasis to specific organs appear to affect prognosis. The 5-year survival rate was 6% and 12% for patients with brain and liver metastases, respectively. This was markedly lower than that in cohorts with bone (25%) and liver (21%) metastasis. Risk factors that significantly influence overall survival were male gender, multiple organ involvement, and brain metastasis. Therefore, we should take into consideration of such discrepancy when making treatment strategies.

**Abstract:**

Introduction—heterogeneity in clinical outcomes and survival was observed in patients with papillary thyroid cancer (PTC) and distant metastases. Here, we investigated the effect of distant metastases sites on survival in PTC patients. Methods—patients with a diagnosis of PTC and known metastases were identified using the Surveillance, Epidemiology, and End Results database (1975–2016). Univariate and multivariate Cox regression analyses were performed to analyze the effect of distant metastases sites on thyroid cancer-specific survival (TCSS) and overall survival (OS). Results—from 89,694 PTC patients, 1819 (2%) developed distant metastasis at the initial diagnosis, of whom 26.3% presented with the multiple-organ disease. The most common metastatic sites were lung (53.4%), followed by bone (28.1%), liver (8.3%), and brain (4.7%). In metastatic patients, thyroid cancer-specific death accounted for 73.2%. Kaplan–Meier curves showed decreased OS in patients with metastases to the brain (median OS = 5 months) and liver (median OS = 6 months) compared to lung (median OS = 10 months) and bone (median OS = 23 months). Moreover, multiple organ metastasis had a higher mortality rate (67.4%) compared to single organ metastasis (51.2%, *p* < 0.001). Using multivariate analysis, risk factors that significantly influence TCSS and OS were male gender (HR = 1.86, 95% CI = 1.17–2.94, *p* < 0.001, and HR = 1.90, 95% CI = 1.40–2.57, *p =* 0.009), higher tumor grade (HR = 7.31, 95% CI = 2.13–25.0, *p* < 0.001 and HR = 4.76, 95% CI = 3.93–5.76, *p* < 0.001), multiple organ involvement (HR = 6.52, 95% CI = 1.50–28.39, *p* = 0.026 and HR = 5.08, 95% CI = 1.21–21.30, *p* = 0.013), and brain metastasis (HR = 1.82, 95% CI = 1.15–2.89, *p* < 0.001 and HR = 4.21, 95% CI = 2.20–8.07, *p* = 0.010). Conclusion—the pattern of distant metastatic organ involvement was associated with variability in OS in PTC. Multi-organ metastasis and brain involvement are associated with lower survival rates in PTC. Knowledge of the patterns of distant metastasis is crucial to personalize the treatment and follow-up strategies.

## 1. Introduction

Thyroid cancers (TC) are the most common endocrine malignant tumors. Their incidence is expected to be 44,280 in 2021, making them the 7th most common cancer in women in the U.S. [1]. Women are three times more commonly affected than men, and the average age at diagnosis is around 40 years [2]. Primary thyroid cancers (TC) are histologically divided into four groups—(I) well-differentiated epithelial thyroid cancers, (II) poorly differentiated epithelial thyroid cancers, (III) medullary thyroid cancers, and (IV) rare thyroid tumors (lymphoma, sarcoma, squamous cell, etc.) [3].

Well-differentiated epithelial thyroid cancer includes either papillary thyroid cancer (PTC), follicular thyroid cancer (FTC), or Hurthle cell thyroid cancer (HCTC) [4]. Papillary carcinoma is the most common histologic type of TC, accounting for approximately 80% to 90% of all newly diagnosed thyroid cancers [5]. PTC arises from the follicular cells of the normal thyroid gland. It is characterized by a papillary growth pattern of tumor cells with distinctive nuclear features as (1) overlapping and enlarged nuclei; (2) pale and optically clear; and (3) nuclear membrane irregularities [5]. Several PTC variants have been identified, presenting different prognosis and outcome [6]. Follicular variants (FV-PTC) represent 25% of PTCs, contain follicular, as well papillary features, and are associated with a favorable prognosis. Inversely, tall-cell variants (TV-PTCs) and diffuse sclerosing variants (DSVs) are aggressive PTCs. TV-PTCs compose 10% of all PTCs and are characterized by cells that are two times taller than in a classical PTC. DSV-PTCs represent 3% of all PTCs and include psammoma bodies and extensive calcification. These aggressive histological subtypes are associated with thyroid invasion, regional and distant metastasis. They present a significant reduction of patient overall survival, with a 5-year survival rate of 87.5% and 80.6% for DSV-PTC and TV-PTC, respectively [7,8,9]. 

Globally, PTC is one of the most treatable cancers in comparison to other differentiated and undifferentiated thyroid malignancies [10], leading to a survival rate of 93% at 10 years [11]. Around 30% to 40% of PTC metastasize to regional lymph nodes [5,12]. However, distant metastasis may occur and accounts for 1–4% of the patients, reducing the survival rates to 24–76% [1,10]. The lungs and the bones are the most common sites for distant metastasis [10,13]. Interestingly, the outcomes drastically decline in patients exhibiting multi-organ metastasis, with a 5-year survival rate of 15.3%, compared to 77.6% for patients with single-organ metastasis [13].

Risk factors for distant metastases include male gender, advanced age [14], histologic grade [15], completeness of surgical resection of the primary tumor [14], extrathyroidal extension, and lymph node metastasis at initial examination [16]. However, some patients with distant metastases achieve a complete remission or very long periods of progression-free survival for many years. Others, however, rapidly progress and die. This variability in clinical and survival outcomes suggests significant heterogeneity in thyroid distant metastatic disease, which might influence their management paradigm and follow-up studies.

Population-based cancer registries provide an excellent opportunity to investigate the relationship between the patterns of distant metastases and prognosis in metastatic cancer. The aims of the current nationwide study were to (1) determine the impact of single and multi-organ distant metastases on survival; (2) assess the differential consequence of specific metastatic sites on survival; and (3) identify factors that predict survival in patients with distant metastasis in a diverse population representing wider geographic regions in the United States.

## 2. Methods and Materials

### 2.1. Data Source

A retrospective cohort study was performed using the Surveillance, Epidemiology, and End Results (SEER) database (https://seer.cancer.gov/about/overview.html: accessed date: 31 January 2021); a population-based clinical oncology registry covering approximately 34.6% of the United States citizens. The SEER data are publicly available, thus the institutional review board approval was exempted, and the patient’s written consent was waived. We signed the Research Data Agreement before this study and got access to the database with the username of 15332-Nov2019.

### 2.2. Cohort Extraction

We extracted PTC patients from the SEER 9 registry (1975–2016) using the SEER∗Stat software (version 8.3.6; Surveillance Research Program, National Cancer Institute, Bethesda, MD, USA; www.seer.cancer.gov/seerstat: accessed date: 31 January 2021) and imported into IBM Statistical Packages of Social Statistics (SPSS) version 27.0 (Armonk, NY, USA; IBM corp.). International Classification of Diseases for Oncology (ICD-O-3) was adopted to identify the cancer site (Thyroid) and histology type (PTC: 8050, 8260, 8340–8344, 8350, 8450–8460). The SEER database included 212,651 PTC patients during the study period. Of these, 89,694 have reported data on metastasis stage at initial diagnosis; either non-metastatic (M0) or metastasis (M1) to liver, lung, bone, brain, or distant lymph nodes. Cases with undetermined TNM stage were deleted. Patients with any other malignancies were excluded. Both adults (>18 years) and pediatric cohorts were included to identify the heterogenous pattern in both age groups.

### 2.3. Variables and Outcomes

The SEER data include demographic and clinical characteristics information, including patient identification, age of diagnosis, year of diagnosis, gender, race/ethnicity, histology type, insurance status, tumor size, regional lymph node (LN) status, distant metastatic site, cause-specific death classification, other cause of death classification, and survival months. Patients were analyzed as two groups based on the number of involved organs into single- and multi-organ distant metastases (SODM and MODM) and categorized according to the site of metastasis into lung, bone, liver, brain, distant LN, and others. Patients with different distant metastatic sites at the initial presentation were compared.

### 2.4. Outcome Measures

The main outcome was overall survival (OS), defined as the time from the date of diagnosis to the date of death or last follow-up. In SEER, reported reasons for death were identified and classified into thyroid cancer, cancer, and non-cancer causes. Thyroid cancer-specific survival (TCSS) was estimated and compared across the study groups. TCSS was defined as a net survival measure representing cancer survival in the absence of other causes of death based on causes of death listed in medical records.

### 2.5. Statistical Analysis

Statistical analysis was conducted using SAS Version 9.4 and SPSS Version 26. Patient and tumor characteristics were compared using the two-sided chi-square test for categorical covariates and Student’s t, One-Way ANOVA, Mann–Whitney U, and Kruskal Wallis tests for numerical covariates. The significance level was set at *p* < 0.05. Kaplan–Meier (KM) survival curves and the log-rank test were used for determining OS and the specific probability of survival for each group of patients. Multivariate Cox Hazards Proportionate Regression analysis model was performed to determine independent prognostic factors of TCSS and OS. Hazard ratios (HRs) and 95% confidence intervals (CIs) were calculated. Concordance or c-statistics were used to assess the predictive accuracy of the models.

## 3. Results 

### 3.1. Characteristics of Metastatic Cohorts

A total of 212,651 thyroid cancer patients (1975–2016) from the SEER database were reviewed and only those with known metastasis were included in the analysis (*n* = 89,694 patients). Baseline characteristics of thyroid cancer patients with and without metastasis are demonstrated in Table 1. The median age of patients with metastasis at initial presentation was 65.38 ± 15.95 years compared to 50.25 ± 15.64 years in non-metastatic cohorts. Distant metastasis was reported in 1819 patients (2.03%) at the time of diagnosis. Lung was the most common site of metastasis, reported in 1290 patients (53.4%), followed by bone metastasis (*n* = 680 patients, 28.1%). The incidence of metastasis to the liver and brain accounted for 8.3% and 4.7%, respectively. Distant lymph node metastasis was found in only 67 patients (2.8%). A total of 1341 metastatic thyroid cancer patients (73.7%) presented with single organ involvement, while 478 patients (26.3%) had MODM at the time of initial diagnosis. Involvement of two, three, and four metastatic sites was observed in 383, 74, and 19 patients, respectively (Figure 1). The median follow-up for the entire patient cohort was 34 months (range 1–83 months). Of 1035 pediatric thyroid cancer patients, only 22 (2.12%) presented with single organ distal metastasis in the liver (14 females and 8 males) and were alive at the end of the follow-up period. Mortality was reported for other reasons in seven non-metastatic pediatric cases.

### 3.2. Impact of Single and Multi-Organ Distant Metastases on Prognosis

Table 2 demonstrates a comparison between patients with single- and multi-organ involvement. No significant difference between the two groups regarding their age (*p* = 0.96), sex (*p* = 0.87), or race (*p* = 0.28) was seen. SODM occurred most commonly in the lung (65.2%), bone (24.8%), and liver (4.8%). The most common site of MODM was the lung (87%), bone (72.6%), and liver (28.5%), followed by the brain (17.8%). Despite MODM, patients had a lower frequency of positive LNs (32%) compared to the SODM group (38.9%), mortality rates were significantly higher in thyroid cancer patients presented with MODM at the initial diagnosis (overall mortality rate—67.4% versus 51.2%, *p* < 0.001; cancer-specific mortality rate: 50.4% versus 36.7%, *p* < 0.001) compared to SODM. One- and five-year OS were 50% and 28% in SODM patients from the diagnosis. These figures were markedly lower (28% and 11%) in MODM patients (*p* < 0.001) (Figure 2). Kaplan–Meier survival curves demonstrated poor survival in MODM cohorts compared to SODM; median estimates of overall survival were 6 months in MODM versus 29 months in SODM (*p* < 0.001), and that of thyroid cancer-specific survival was 12.0 months in MODM versus 70.0 months in SODM (*p* < 0.001).

### 3.3. Impact of the Site of Metastasis at Presentation on Prognosis

A comparison between patients with distant metastasis at various sites is illustrated in Table 3. There was no significant difference in age (*p* = 0.54) and gender (*p* = 0.51) between the groups. However, a higher frequency of bone metastasis was observed in black adults while Asian/Pacific Islanders were more prone to brain metastasis (*p* = 0.001). In patients with a single metastatic site, lung (*n* = 874, 67.8%) was the preferential site of distant metastasis in thyroid cancer patients, followed by bone metastasis (*n* = 333, 49.0%). In contrast, the brain was more likely to present with concomitant metastatic sites, such as the lung (*n* = 37, 2%) and liver (*n* = 29, 1.6%). Death was the highest in patients with brain metastasis (72.6%), followed by and liver (68%), lung (61.2%), and bone (52.4%) metastases. In contrast, mortality was reported in 26.9% of patients with distant LN metastasis (*p* < 0.001). In metastatic patients, thyroid cancer-specific death accounted for 73.2%. Mortality due to non-cancer causes was reported in 19.8% mainly due to respiratory disorders (5.95%), heart diseases (3.07%), and septicemia (1.39%).

As depicted in Figure 3, Kaplan–Meier curves showed unfavorable prognosis in patients with metastasis to the brain (median overall survival = 5 months, 95% CI = 2.4–7.5) and liver (median = 6 months, 95% CI = 3.8–8.1) compared to bone (median = 23 months, 95% CI = 7.8–12.1) and lung (median = 10 months, 95% CI = 15.7–30.2). The 5-year survival of patients with brain (6%) and liver (12%) metastasis was markedly less than that in cohorts with the bone (25%) and liver (21%) metastasis.

Using multivariate analysis, risk factors that significantly influence overall survival (Figure 4A) and thyroid cancer-specific survival (Figure 4B) were male gender (OS: HR = 1.90, 95% CI = 1.40–2.57, *p* < 0.001; and TCSS: HR = 1.86, 95% CI = 1.17–2.94, *p* = 0.009), higher tumor grade (OS: HR = 4.76, 95% CI = 3.93–5.76, *p* < 0.001; and TCSS: HR = 7.31, 95% CI = 2.13–25.0, *p* < 0.001), and metastasis (OS: HR = 4.09, 95% CI = 2.28–7.33, *p* < 0.001; and TCSS: HR = 4.29, 95% CI = 2.23–8.27, *p* < 0.001). Subgroup analysis by number and site of metastasis revealed a five- to six-fold higher risk of mortality in the presence of multiple organ involvement (OS: HR = 5.08, 95% CI = 1.21–21.30, *p* < 0.001; and TCSS: HR = 6.52, 95% CI = 1.50–28.39, *p* < 0.001), and patients were around two- to four-fold more likely to die with brain metastasis (OS: HR = 4.21, 95% CI = 2.20–8.07, *p* < 0.001; and TCSS: HR = 1.82, 95% CI = 1.15–2.89, *p* = 0.010) (Figure 4C,D). In contrast, after adjustment, age at diagnosis, lymph node involvement, and tumor size at presentation did not show a significant influence on mortality. In addition, there was no differential survival outcomes in patients with other metastatic sites as lung, bone, liver, or distant lymph nodes compared to non-metastatic PTC cohorts (Figure 4).

## 4. Discussion

The prognostic value of each site in metastatic TC remains controversial. Population-based cancer registries provide an excellent opportunity to investigate the relationship between the patterns of distant metastases and prognosis in metastatic disease [17,18]. In this study, we investigated whether the survival of thyroid cancer patients was differentially affected by the location of distant metastases. Our results revealed that the pattern of organ-specific metastases has different prognostic values in PTC.

In the SEER database, distant metastasis was reported in nearly 2% of thyroid cancer patients. In previous publications, spread of thyroid cancer outside the neck was previously reported to be rare at the time of initial presentation, occurring in between 1.2% and 13% of patients [19,20,21,22,23,24]. The most common metastatic sites in our patients were lung (53.4%), followed by bone (28.1%), liver (8.3%), and brain (4.7%), with multiple organ involvement accounting for 26% of the cohort. This was consistent with several previous studies [24,25]. Borschitz and his colleagues [26] revealed lung and bone to be the preferential sites for thyroid cancer metastasis. Similarly, the article published by Ding et al. [27] reported the lungs as the most common site of distant metastasis in differentiated thyroid cancer, representing 47.7% of the population, followed by bone metastasis (24.9%). However, liver (19.5%) and brain (7.9%) metastases were double the prevalence found in the current data of the SEER 9 registry and exhibited less frequency of multiple organ metastases (19%), as was reported in [27]. Ding et al. used the SEER 18 registry (2010–2016) which covered both papillary and follicular TC patients from 18 geographical regions in the United States [27]. In contrast to our findings, the study of Chen et al. [19] showed higher frequency of brain metastasis than liver. Other sites such as liver, brain, skin, skeletal muscle, ovaries, oropharynx, submandibular gland, sphenoidal sinus, adrenal gland, and pancreas have been reported [17,26,28].

Although PTC is a disease with a generally good outcome, DTC patients presenting with distant metastasis have less favorable outcomes. In the current study, metastatic patients exhibited inflated mortality rate (55.5%) compared to the non-metastatic group (4.3%). Furthermore, multiple organ metastasis was associated with higher frequency of death (50.4%) versus single organ involvement (36.7%). Despite being the least common, univariate analysis showed higher mortality rates in thyroid cancer cohorts presenting with brain (72.6%) and liver (68%) metastases compared to other groups. In contrast, those who developed distant LN metastasis at presentation showed the best survival rate. In our study, patients with spread to the lung and bone were associated with longer TCSS and OS, whereas brain and liver metastasis at diagnosis was associated with relatively shorter survival. Our findings indicated that the 5-years survival of patients with the bone (25%) and liver (21%) metastases was markedly higher than that in cohorts with brain (6%) and liver (12%) metastases. Consistent with our results, Wang et al. [13] reported that the 5-year survival rate in patients with metastasis limited to one organ was 77.6%, while that in patients who develop second organ involvement was as low as 15.3%. Ding et al. [27] demonstrated that the 3-year thyroid cancer-specific survival rate of bone metastasis from DTC was 50.4%, lung metastasis was 45.9%, liver metastasis was 40.9%, and brain metastasis was 28.7%. However, Sampson et al. [25] reported the 3-year survival of single organ metastatic patients to be 77% in the presence of lung metastasis, significantly higher than bone metastasis (56%). These survival variations according to the location of metastasis could be probably due to the differences in biological heterogeneity of tumor clones spreading to each site, as well as treatment strategies and accessibility for tumor eradication.

In the current study, after adjustment, only patients with brain metastasis or multi-organ involvement were associated with a significantly less favorable prognosis, whereas other metastatic sites did not show an impact on survival. Brain metastasis was also reported previously in other types of cancer. Lung (16.3%), renal (9.8%), melanoma (7.4%), breast (5.0%), and colorectal (1.2%) cancers take the lead, respectively, in forming brain metastases [29]. While in children, sarcomas, neuroblastoma, and germ cell tumors are the most common brain metastasis sources [30,31,32]. When origin of brain metastases is compared, PTC brain metastases are the least common among other types of cancer [33]. However, undiagnosed asymptomatic brain metastasis was reported. An autopsy series of DTC patients showed that up to 20% of the group has central nervous system (CNS) involvement [34]. Al-Dhahri et al. underlined that brain metastasis occurs more frequently in the cerebral hemispheres, and other sites of intracranial metastasis are the cerebellum, brainstem and pituitary [16]. Among these, cranial metastases involving the skull only were associated with a better outcome [35], while those involving the brainstem and cranial neuropathy could lead to unfavorable outcomes [36,37]. In addition, patients with multiple cranial metastases seemed to suffer a worse outcome than patients with a single metastasis [35].

Brain metastases by their very nature represent an extreme and immediate threat to patients. The assessment tool, Karnofsky Performance Scale, was proposed to be an indicator for predicting survival and functional impairment [38]. Surveillance imaging of the head during perioperative management or follow-up for PTC patients is not routinely considered unless there is clinical suspicion following neurological symptoms [34,39]. A significant survival advantage for definitive local primary tumor surgery was observed in DTC patients with single- or multi-organ metastasis, except for patients with only brain metastasis where surgery produced no survival benefit over non-operative management [27]. In addition to surgery, radiotherapy or radioactive iodine (RAI) therapy have been widely adopted to treat DTC patients with distant metastases. However, the uptake of RAI by cranial metastatic lesions is quite low (0–25% of cases) [34,40,41,42], which might be due to the reduced expression of the sodium iodide symporter (NIS) in metastatic lesions [43]. The treatment of brain metastatic lesions is further made challenging by the generally poor permeability of the blood-brain barrier to modern chemotherapeutics [44]. Surgery is the most effective method for patients with solitary CNS metastases [45].The National Comprehensive Cancer Network (NCCN) guideline recommended that surgical resection followed by whole brain radiation therapy (WBRT) or stereotactic radiation therapy (SRS) plus WBRT was appropriate for patients who had stable systemic disease or were newly diagnosed, while WBRT or SRS was advisable for patients who had multiple (>3) metastatic lesions [46]. Therefore, knowledge of the patterns of distant metastasis is crucial to personalize the treatment and follow-up strategies. Considering the grave course of brain metastasis, the early detection and aggressive treatment of PTC patients is recommended. Also, discovering novel liquid biopsy markers for brain metastasis and innovative targeted therapy and immunotherapy might achieve better survival.

Despite the fact that the risk of metastasis has been traditionally predicted by prognostic factors such as tumor size, axillary lymph node status, and histological grade, tumor cells that are “peeled off” to adjacent blood microvasculature circulate to finally reside in a distant site. Preferential metastasis to specific organs, “organotypic metastasis,” is augmented by crosstalk between the primary cancer cells and host organs and requires modifications of the microenvironment niche of target organs prior to colonization of tumor cells [47,48]. Unravelling the molecular stimulus that govern the cancer cell–chemokine interactions in the setting of brain metastases is crucial for the development of more accurate diagnostics and efficacious therapies. Emerging studies are exploring the role of brain-derived extracellular vesicles in maintaining long-distance communication with the primary cancer cells and provide the signals for the migration of metastatic cells to the brain [49,50]. Tumor cells harboring specific genetic alterations can drive differential clinical outcomes. The *TERT* gene mutation was frequently observed in thyroid cancer brain metastases [51]. Additionally, *BRAF* mutations have been linked to the increased risk of brain metastasis at first diagnosis in such tumors [52]. Further studies are needed to define the role of genomic mutations in brain metastases. Despite the significant progress made over the last decades, our understanding of organotropism and metastatic progression in thyroid cancer patients remains limited. Discovering biomarkers that could predict and prevent organ-specific colonization in thyroid cancer patients may help to determine the appropriate strategy for follow-up and will open new avenues towards personalized medicine. Our work suggests it will be crucial to identify patients in whom brain metastases may evolve, with other sites being more common but having less impact on OS.

Based on the literature, thyroid cancer represents only 1% to 4% of all pediatric cancers [53,54,55]. At diagnosis, children and adolescents with DTC usually present as a more aggressive disease than in the adult population, with a higher percentage of LN metastasis and extrathyroidal extension [55]. Compared to adults, pediatric TC have higher rates of metastases and recurrence [56]. However, patients in this age group often have a favorable survival with uncommon disease-specific death, even in cases of advanced disease [57,58]. Furthermore, childhood DTC with distant metastasis persists in most patients despite multiple courses of RAI [58,59,60]. In our results, pediatric cohorts represented 1.15% of the study population. Of these, 2.12% presented with liver metastasis at the initial time of diagnosis. Within the pediatric group, reported mortality was caused by other causes rather than cancer. In contrast to our findings, higher rates of metastasis of up to 13.3% were reported in a few studies [61,62], and the lungs were shown as the primary site of distant metastasis in several publications, accounting for up to 25% of childhood DTC patients [55,59,63,64,65]. Whether there was underdiagnosis of asymptomatic distant metastasis at other sites or there is a preferential chemotaxis to the liver specifically in our cohorts needs further investigation. Genomic mutation and selection of malignant tumor cells may cause haphazard dislodging and emigration of cells via lymphatics and/or the blood to settle down in any distant organs [66]. However, other theories suggest the presence of tissue-specific adhesion molecules on endothelial cells that selectively attract circulating metastatic cells and form a premetastatic nucleus [67]. Chemokines can regulate organ predilection of metastasis. In an animal model, the chemokine receptor CCR6 was upregulated in small liver metastases of thyroid carcinoma [68], highlighting its putative rule in early detection of pediatric liver metastasis. These results supported that individualized treatment decisions for primary tumor surgery of metastatic thyroid carcinoma patients should be tailored based on the affected locations.

Some limitations need to be addressed. First, the SEER database only included five specific sites of distant metastases at the initial diagnosis, and we could not obtain further details concerning the development of further metastasis after initial diagnosis to evaluate the dynamic metastatic landscape over time and identify factors that predict SODM progressing to MODM. In addition, there was a lack of details in the SEER database concerning targeted therapy and specific treatment for metastatic organs to assess their association with survival and clinical outcomes in different organ-specific metastatic patterns. Finally, pediatric patients represented around one thousand patients with very few metastasis cases in the liver. Further stratification and analysis as a separate disease entity was challenging. Further studies of favorable sites of metastasis in pediatric cohorts are warranted.

## 5. Conclusions

Taken together, our large population-based study suggests that in papillary thyroid carcinoma patients with distant metastases, those with brain and multi-organ metastases have the poorest survival. Therefore, we should take into consideration such discrepancy when making treatment strategies.

## Figures and Tables

**Figure 1 cancers-13-01625-f001:**
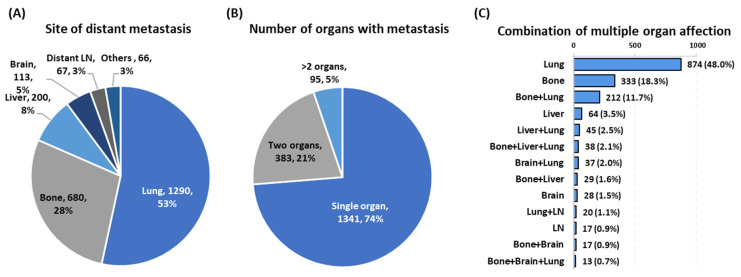
The site and the number of metastatic organs. Data are presented as the number of patients (percentage). (**A**) Frequency of patients according to the site of distant metastasis. (**B**) Frequency of patients according to the number of organs involved. (**C**) Frequency of patients according to the combination of metastases sites per patient.

**Figure 2 cancers-13-01625-f002:**
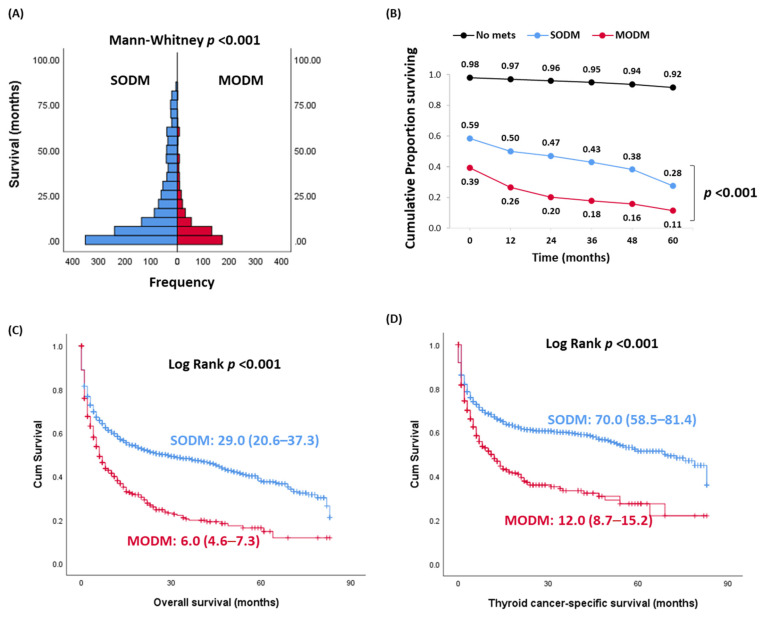
Differential impact of single- and multi-organ distant metastases on survival. SODM: single organ distant metastasis, MODM: multiple organ distant metastasis. (**A**) Histogram comparing the survival in SODM and MODM groups. Mann–Whitney U test was applied. (**B**–**D**) Kaplan–Meier survival curves comparing between SODM and MODM groups for overall survival and thyroid cancer-specific survival. Log Rank test was used. Overall survival was defined as the time between diagnosis and death from any cause, and thyroid cancer specific survival was defined as the time between diagnosis and death from DTC.

**Figure 3 cancers-13-01625-f003:**
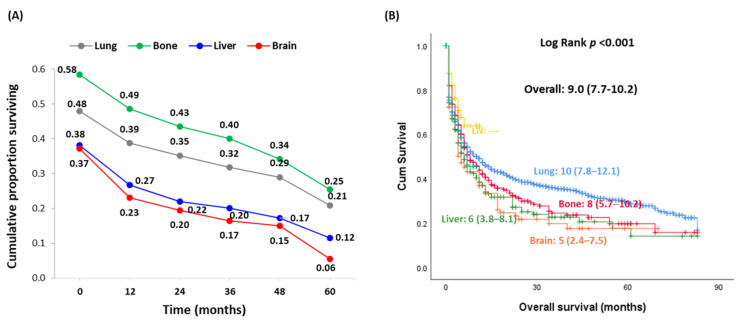
Impact of the site of metastasis at presentation on overall survival. (**A**) Five-year overall survival. (**B**) Kaplan–Meier survival curve comparing between patients presenting with metastasis at different sites. Log Rank test was used.

**Figure 4 cancers-13-01625-f004:**
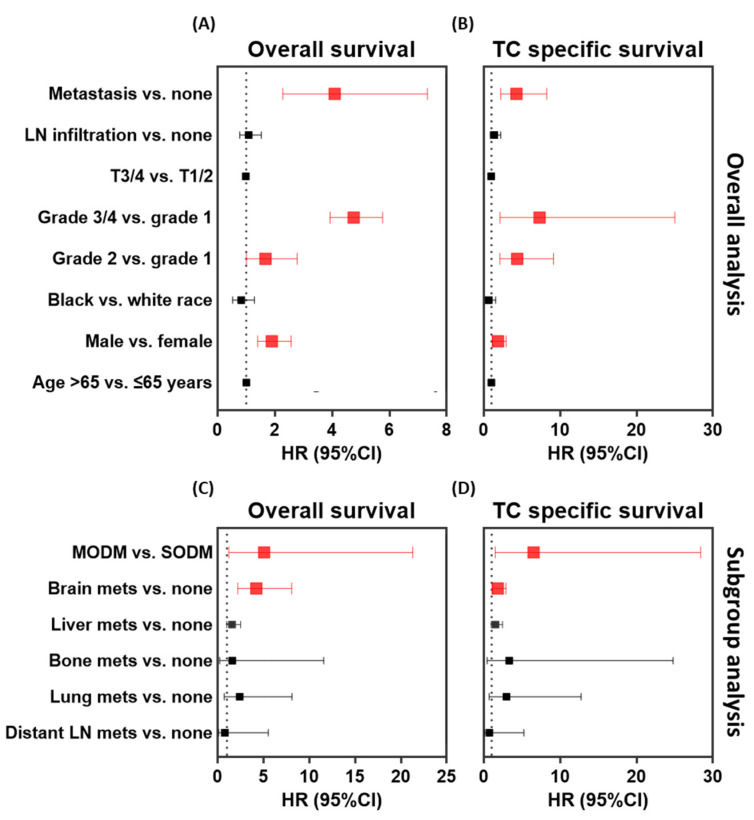
Risk of mortality in patients with metastasis compared to non-metastatic patients. (**A**) Overall analysis for identifying risk factors of overall survival. (**B**) Overall analysis for identifying risk factors for thyroid cancer-specific survival. (**C**) Subgroup analysis for identifying risk factors of overall survival. (**D**) Subgroup analysis for identifying risk factors for thyroid cancer specific survival. Multivariate cox regression models were employed on the whole study population either with or without metastasis. In overall analysis, risk in the presence of metastasis was first performed adjusted with other variables. Next, iterations were performed after replacing the metastasis variable with each other variable listed in the sub-analysis section. Metastasis in each site is compared to all cohorts without metastasis. HR: hazard ratio, 95% CI: confidence interval, SODM: single-organ distal metastasis, MODM: multi-organ distant metastasis, vs: versus, mets: metastasis.

**Table 1 cancers-13-01625-t001:** Baseline characteristics and demographics of patients.

Characteristics	Level	Total Population (*n* = 89,694)	No Metastasis (*n* = 87,875)	Metastasis (*n* = 1819)	*p* Value
Age, years	Mean ± SD	65.18 ± 14.94	50.25 ± 15.64	65.38 ± 15.95	<0.001
	<18 years	1035 (1.2)	1013 (1.2)	22 (1.2)	<0.001
	18–45 years	33,088 (36.9)	32,926 (37.5)	162 (8.9)	
	46–65 years	39,198 (43.7)	38,558 (43.9)	640 (35.2)	
	>65 years	16,373 (18.3)	15,378 (17.5)	995 (54.7)	
Gender	Female	67,635 (75.4)	66,647 (75.8)	988 (54.3)	<0.001
	Male	22,059 (24.6)	21,228 (24.2)	831 (45.7)	
Race	White	71,973 (80.2)	70,635 (80.4)	1338 (73.6)	<0.001
	Black	6356 (7.1)	6157 (7)	199 (10.9)	
	Asian/Pacific Islander	9408 (10.5)	9143 (10.4)	265 (14.6)	
	American Indian/Alaska	609 (0.7)	597 (0.7)	12 (0.7)	
	Unknown	1348 (1.5)	1343 (1.5)	5 (0.3)	
Insurance status	Not insured	2194 (2.4)	2142 (2.4)	52 (2.9)	<0.001
	Private	77,827 (86.8)	76,390 (86.9)	1437 (79)	
	Medicaid/Medicare	9673 (10.8)	9343 (10.6)	330 (18.1)	
Number of metastatic sites	0	87,875 (98.0)	87,875 (100)	---	NA
	1	1341 (1.5)	---	1341 (73.7)	
	2	383 (0.4)	---	383 (21.1)	
	3+	95 (0.1)	---	95 (5.2)	
Vital status	Alive	84,937 (94.7)	84,127 (95.7)	810 (44.5)	<0.001
	Dead	4757 (5.3)	3748 (4.3)	1009 (55.5)	

Data are shown as number and percentage. Chi-square and Student’s *t*-tests were used. *p*-value was considered significant at values less than 0.05.

**Table 2 cancers-13-01625-t002:** Comparison between single- and multi-organ distant metastases.

Characteristics		SODM (*n* = 1341)	MODM (*n* = 478)	*p* Value
Demographic Data				
Age, years	Mean ± SD	65.37 ± 16.6	65.4 ± 13.9	0.96
Sex	Female	730 (54.4)	258 (54)	0.87
	Male	611 (45.6)	220 (46)	
Race	White	1002 (74.7)	336 (70.3)	0.28
	Black	138 (10.3)	61 (12.8)	
	Asian/Pacific Islander	188 (14)	77 (16.1)	
	American Indian/Alaska	10 (0.7)	2 (0.4)	
Cancer characteristics				
Tumor size, mm	Median (Quartiles)	13 (4–25)	18 (7–27)	0.42
	Mean ± SD	16.7 ± 14.9	20.7 ± 16.8	0.28
Regional LN	N0	819 (61.1)	325 (68)	0.008
	N1	522 (38.9)	153 (32)	
Outcomes				
Overall survival, years	Median (Quartiles)	6 (2–10)	3 (1–6)	<0.001
Thyroid cancer-specific death, %	Thyroid cancer	492 (36.7)	241 (50.4)	0.002

Data are shown as number and percentage, mean and standard deviation, or median and quartile. SODM: single-organ distal metastasis. MODM: multi-organ distant metastasis. Chi-square, Mann–Whitney U, and Student’s *t* tests were used. *p*-value was considered significant at values less than 0.05.

**Table 3 cancers-13-01625-t003:** Patient demographic and clinicopathological characteristics according to site of the metastasis.

Haracteristics		Lung (*n* = 1290)	Bone (*n* = 680)	Liver (*n* = 200)	Brain (*n* = 113)	LN (*n* = 67)	Others (*n* = 66)	*p* Value
Demographic Data								
Age, years	Mean ± SD	65.61 ± 16.45	65.61 ± 13.25	65.83 ± 15.11	62.87 ± 14.19	63.82 ± 16.66	65.58 ± 14.51	0.54
Sex	Female	695 (53.9)	370 (54.4)	111 (55.5)	70 (61.9)	36 (53.7)	41 (62.1)	0.51
	Male	595 (46.1)	310 (45.6)	89 (44.5)	43 (38.1)	31 (46.3)	25 (37.9)	
Race	White	960 (74.4)	468 (68.8)	159 (79.5)	77 (68.1)	52 (77.6)	53 (80.3)	0.001
	Black	129 (10)	105 (15.4)	20 (10)	8 (7.1)	4 (6)	6 (9.1)	
	Asian/Pacific Islander	185 (14.3)	103 (15.1)	20 (10)	28 (24.8)	10 (14.9)	7 (10.6)	
Cancer characteristics								
Tumor size, mm	Mean ± SD	17.48 ± 14.48	13.25 ± 11.25	17.91 ± 10.49	28.00 ± 25.96	22.37 ± 21.31	16.64 ± 10.10	0.94
Regional LN	N0	776 (60.2)	324 (47.6)	125 (62.5)	76 (67.3)	39 (58.2)	42 (63.6)	<0.001
	N1	514 (39.8)	356 (52.4)	75 (37.5)	37 (32.7)	28 (41.8)	24 (36.4)	
Number of metastatic organs	One	874 (67.8)	333 (49.0)	64 (32)	28 (24.8)	17 (25.4)	25 (37.9)	<0.001
	Two	324 (25.1)	264 (38.8)	76 (38)	58 (51.3)	26 (38.8)	18 (27.3)	
	≥Three	92 (7.1)	83 (12.2)	60 (30)	27 (23.9)	24 (35.8)	23 (34.8)	
Survival								
Vital status	Alive	500 (38.8)	324 (47.6)	64 (32)	31 (27.4)	49 (73.1)	40 (60.6)	<0.001
	Dead	790 (61.2)	356 (52.4)	136 (68)	82 (72.6)	18 (26.9)	26 (39.4)	
OS, years	Mean ± SD	4.10 ± 3.28	4.88 ± 2.53	3.64 ± 3.29	3.50 ± 3.86	4.63 ± 3.74	4.09 ± 2.88	<0.001
Cause of death, %	TC	587 (74.3)	246 (69.1)	105 (77.2)	61 (74.4)	11 (61.1)	20 (76.9)	0.40
	Other cancers	53 (6.7)	33 (9.3)	7 (5.1)	2 (2.4)	2 (11.1)	2 (7.7)	
	Non-cancer	150 (19)	77 (21.6)	24 (17.6)	19 (23.2)	5 (27.8)	4 (15.4)	

Data are shown as number and percentage, mean and standard deviation, or median and quartile. LN: lymph node, OS: Overall survival, TC: thyroid cancer. Chi-square, Kruskal–Wallis, and One-way ANOVA *t*-tests were used followed by the Tukey test for multiple comparisons. *p*-value < 0.05 was considered significant.

## Data Availability

Publicly available datasets were analyzed in this study. This data can be found here: (https://seer.cancer.gov/, (accessed on 31 January 2021)).

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
