# Peer review of "Site-Specific Metastasis and Survival in Papillary Thyroid Cancer: The Importance of Brain and Multi-Organ Disease"

_cancers, 2021, doi:10.3390/cancers13071625_

Round 1
Reviewer 1 Report
Ref: Cancers- 1116608
Site-specific metastasis and survival in papillary thyroid cancer: the importance of brain and multi-organ disease
Journal: Cancers-MDPI
The manuscript entitled: “Site-specific metastasis and survival in papillary thyroid cancer: the importance of brain and multi-organ disease” by Toraih et al. is an article that describes the pattern of distant metastatic organ involvement associated with variability in overall survival of papillary thyroid cancer. The study fits within the scope of the journal. This manuscript needs major revisions before the acceptance for publication in the Cancers Journal. Please find below the comments-suggestions and major revisions that will help the authors improve the current version of this manuscript:
Major/minor comments:
Abstract: lines 33-35: the conclusion needs to be re-written to reflect the overall study.
Introduction
- line 46-48: “However, distant metastasis may occur and accounts for 1-4% of the patients. The lungs and the bones are the most common sites for distant metastasis”: since the % of metastasis in patients for thyroid cancer is 4%, please explain within a few lines the significance of further study the metastatic cascade for this kind of cancer.
-In general, the authors could improve the introduction by the addition of some extra paragraphs/references (i.e expanding on the major problem of cancer metastasis; explaining why this study could provide important information for thyroid cancer etc)
-line 44-45: “PTC is one of the most treatable cancers, leading to a survival rate of 93% at 10 years”: please justify the choice of only PTC and not other mentioned types of thyroid cancers.
Methods
-line 70: “Both adults (>18 years) and pediatric cohorts were included”: please discuss the fact that there may be differences in the clinical outcome/metastatic potential between adults and pediatric cohorts and how you could deal with this.
-lines 74-75: “categorized according to the site of metastasis into lung, bone, liver, brain, distant lymph node (LN), and others”: please discuss more the significance of these specific metastatic sites for thyroid cancer compared to other studies (and possibly other types of cancers)
Results
-Figure 2,3: Kaplan-Meier survival curves: it would be useful to clearly represent the progression-free survival (PFS) as well.
-Table 4: It would be useful to represent the full analysis of the Multivariate cox regression models as panels in one figure and explain better the given information (in the results section).
Discussion/Conclusion
-Discussion and conclusion could be more critical and concise.
-Conclusion should be re-written to better reflect the overall study.
-It would be useful for the readers to up-date the literature with the addition of more references in the discussion (and introduction) parts.
-The discussion part could be improved and enriched with more literature (see below):
-Lines 195-196: “The most common metastatic sites were lung, followed by bone, liver, and brain”: please expand and explain more this result in comparison with the statement that “In the univariate analysis, mortality rates were higher in thyroid cancer cohorts with brain and liver metastases”(lines 202-203).
-Line 205-207: “In the multivariate Cox analysis, only patients with brain metastasis or multiple organ involvement were associated with a significantly less favorable prognosis, whereas other metastatic sites did not show an impact on OS”: please elaborate with more discussion and literature focusing on brain cancer metastasis and the impact on OS as well as the possible link with treatments.
-Lines 225-230: It would be useful to mention and compare other types of cancers that are susceptible to develop brain cancer metastasis with the chosen thyroid cancer (including references).
Reviewer 2 Report
This analysis provides interesting results concerning the outcome of metastatic PTC.
Some clarifications are needed:
In the abstract "Moreover, multiple organ metastasis had decreased OS (67.4%) compared to single organ metastasis (51.2%, p<0.001)". maybe you meant 51.2% MODM vs 67.4% SODM
In introduction "and extrathyroidal invasion at initial examination" refers to Extrathyroidal extension or lymphnodal or other organs metastases?
The author stated "Of 1,035 pediatric thyroid cancer patients, only 22 developed single organ distal metastasis in the liver" do you confirm that all metastatic pediatric patients presented with liver metastases at diagnosis?
page 4 line 124 "lower frequency of positive LN infiltration" I would delete the word infiltration
page 4 line 128 " from the time of first distant metastasis" it is misleading, better to say from the diagnosis since those are all patients with metastatic disease at the diagnosis
Figure 2, cancer specific survival and thyroid cancer-specific survival, can you explain in the methods the differences of these two analysis? it would be better to delete the curve of cancer specific survival.
Round 2
Reviewer 1 Report
Ref: Cancers-1116608-R1
Site-specific metastasis and survival in papillary thyroid cancer: the importance of brain and multi-organ disease
Journal: Cancers-MDPI
The manuscript entitled: “Site-specific metastasis and survival in papillary thyroid cancer: the importance of brain and multi-organ disease” by Toraih et al. is an interesting article that describes the pattern of distant metastatic organ involvement, associated with variability in overall survival in papillary thyroid cancer (PTC). All the comments and suggestions have been addressed by the authors adequately in the revised manuscript. I believe that now the manuscript has been improved and can be considered for acceptance for publication on this journal in the current version.